# What Came First, Mania or Depression? Polarity at Onset in Bipolar I and II: Temperament and Clinical Course

**DOI:** 10.3390/brainsci14010017

**Published:** 2023-12-23

**Authors:** Delfina Janiri, Alessio Simonetti, Lorenzo Moccia, Daniele Hirsch, Silvia Montanari, Marianna Mazza, Marco Di Nicola, Georgios D. Kotzalidis, Gabriele Sani

**Affiliations:** Department of Neuroscience, Section of Psychiatry, Fondazione Policlinico Universitario Agostino Gemelli IRCCS, Università Cattolica del Sacro Cuore, Largo Francesco Vito 1, 00168 Rome, Italy; alessio.simonetti@policlinicogemelli.it (A.S.); lorenzo.moccia@unicatt.it (L.M.); danielehirsch.com@gmail.com (D.H.); silvia.montanari@yahoo.com (S.M.); marianna.mazza@policlinicogemelli.it (M.M.); marco.dinicola@policlinicogemelli.it (M.D.N.); giorgio.kotzalidis@gmail.com (G.D.K.); gabriele.sani@unicatt.it (G.S.)

**Keywords:** bipolar disorder, polarity at onset, affective temperaments, manic/hypomanic polarity, depressive polarity

## Abstract

(1) Background: Bipolar disorder (BD) is divided into type I (BD-I) and type II (BD-II). Polarity at onset (PO) is a proposal to specify the clinical course of BD, based on the type of the first episode at disorder onset—depressive (D-PO) or manic (M-PO). At the same time, affective temperaments represent preexisting variants of the spectrum of affective disorders. Our objectives were to investigate the hypothesis that temperament may exert an influence on PO, and that this factor can serve as an indicator of the forthcoming course of the disorder, carrying significant therapeutic implications. (2) Methods: We included 191 patients with BD and examined clinical variables and temperament; the latter was assessed using the short version of the Temperament Evaluation of Memphis, Pisa, Paris, and San Diego—Auto-questionnaire (TEMPS-A-39-SV). We tested the associations between these variables and PO using standard univariate/bivariate methods followed by multivariate logistic regression models. (3) Results: 52.9% of the sample had D-PO and 47.1% had M-PO. D-PO and M-PO patients scored higher for dysthymic and hyperthymic temperaments, respectively (*p* < 0.001). Also, they differed in BD subtypes, age at first affective episode, illness duration, number of depressive episodes, seasonality, suicide risk, substance use, lithium, and benzodiazepine use (*p* < 0.05). Only BD-II and age at first depressive episode were predictors of D-PO, whereas BD-I, age at first manic/hypomanic episode, and hyperthymic temperament were predictors of M-PO (*p* < 0.01). (4) Conclusions: Our findings point to the importance of carefully assessing temperament and PO in patients with BD, to better predict the clinical course and tailor therapeutic interventions to individual patients’ needs.

## 1. Introduction

Bipolar disorders (BD) are highly heterogeneous. The heterogeneity of BD has been demonstrated at both the clinical [1] and biological levels [2,3]; interindividual differences have been highlighted in patients with all types of BD. Accounting for heterogeneity allows us to stratify BD into more homogeneous subtypes and to enforce specific interventions and adapt treatment strategies to individual patient needs [4].

Polarity at disease onset (PO) has been suggested to underpin part of the clinical heterogeneity of BD. It refers to the predominant mood state at the beginning of the illness. It serves as a course specifier, indicating whether the initial episodes of the disorder were characterized by mania, hypomania (elevated mood), or depression. Previous studies showed that identifying the polarity at onset is valuable for understanding the trajectory and clinical expression of BD [5,6,7,8]. For instance, individuals with a depressive onset may experience a different course and symptomatology compared to those with a manic or hypomanic onset. Accordingly, the type of episode (depressive or manic) that first occurs in the course of BD may distinguish groups of individuals who differ in the clinical outcome of the illness [5,6,7,8]. In addition, polarity at disease onset has also been shown to be a familial trait in BD, with concordance in kin pairs [9]. In a study involving 971 subjects from 507 families identified through sibling pairs with “type-I BD”, the authors observed pairs that were concordant for mania at onset. This concordance occurred significantly more frequently than would be expected by chance [9].

Most past studies have explored the association between polarity at disease onset and clinical course, typically in samples comprising individuals diagnosed with “type-I BD” [5,6,7]. Several lifetime clinical features differed between patients with “type-I BD” according to the type of episode occurring at the onset of the illness. At the beginning of this millennium, Perugi et al. [6] examined polarity at disease onset in a large sample of individuals with “type-I BD”. They found that depressive polarity at onset was the most common, accounting for 50% of cases. Overall, the trend in episode polarity over time mirrored the polarity observed at the beginning of the illness. Accordingly, depressive onset was associated with more depressive episodes than manic or hypomanic episodes. Individuals with a depressive onset also exhibited higher rates of rapid cycling and suicide attempts, yet were significantly less prone to the development of psychotic symptoms. The authors supported the existence of distinctive longitudinal patterns in “type-I BD” based on their findings, and these patterns appeared to be associated with the polarity observed at the disease onset. Some years later, Perlis et al. [7] specifically explored the hypothesis that the initial occurrence of a depressive episode rather than a manic episode in BD might indicate a subsequent course characterized by a higher burden of depressive symptoms. The study involved the retrospective analysis of data on the first mood episode polarity from 704 individuals with “type-I BD”. Depressive onset was found to be more prevalent among women and those with an earlier onset of the illness. It was significantly linked to a higher number of lifetime depressive episodes and a greater proportion of time spent experiencing depression and anxiety in the year leading up to the study. The authors concluded that the polarity of the first mood episode could be a valuable factor in identifying subsets of patients with BD at risk for a more chronic course. Finally, Forty et al. [5] examined polarity at disease onset in a large, well-characterized sample of patients with “type-I BD”. The authors confirmed that the lifetime clinical features significantly associated with a depressive episode at the beginning of the illness were an earlier age at onset, more frequent and more severe depressive episodes, and less prominent lifetime psychotic features.

Only one study examined the clinical profile of both BD subtypes using polarity at disease onset [8] and corroborated the clinical validity of this construct in predicting clinical outcome. Eduard Vieta’s group in Barcelona conducted a 10-year follow-up prospective study, gathering data from 300 individuals diagnosed with “type-I BD” and “type-II BD”. The sample was divided into two groups based on the polarity of the onset episode. The study revealed that 67% of the patients had a depressive onset. Those with a depressive onset exhibited a more chronic course compared to those with a manic onset, showing a higher number of total episodes and a longer duration of illness. Additionally, patients with a depressive onset experienced a greater frequency of depressive episodes, whereas those with a manic onset had more manic episodes. Depressive onset patients had a higher incidence of suicide attempts, a later onset of illness, fewer hospitalizations, and were less likely to develop psychotic symptoms. Interestingly, the authors discovered that the onset of depression was more common among patients diagnosed with “type-II BD”.

Affective temperaments, as conceptualized by Agop Akiskal, represent enduring and stable patterns of emotional reactivity and regulation that influence an individual’s overall mood disposition. Akiskal’s model, derived from the broader field of temperament in psychology, focuses specifically on emotional traits related to mood disorders [10]. These temperaments, including hyperthymic, depressive, cyclothymic, irritable, and anxious, provide a framework for understanding the predisposition to various affective states. For example, hyperthymic individuals may exhibit a persistent elevation in mood, while those with a depressive temperament may be prone to sustained periods of low mood. Akiskal’s work has significantly contributed to the nuanced understanding of mood disorders, shedding light on the diverse ways individuals experience and express emotions, ultimately aiding in the identification, classification, and treatment of mood-related conditions. The concept of affective temperaments enriches the exploration of the interplay between inherent emotional traits and the development of mood disorders [11].

Interestingly, premorbid temperament types may play an important role in the clinical development of mood episodes, including the direction of affective episodes. As early as 1992, Akiskal proposed that the temperament’s polarity appears to shape the phenomenology of affective episodes differentially [12]. Other investigations confirmed this initial speculation [13,14]. In particular, one study concentrated on the connection between temperament and the psychopathological features of mood episodes. It underscored that when temperaments align with the affective episode, the characteristics of the mood episode are more likely to display the same emotional tone [15]. For instance, euphoric mania may be more prevalent in individuals with a predominant hyperthymic temperament, while depressive characteristics may be more common in those with a prevalent depressive temperament [15]. Findings are consistent with the original hypothesis that the presence of different affective temperaments might influence the phenomenology of affective episodes.

Only one study investigated the relationship between polarity at onset and premorbid affective temperament. In a large sample of patients with “type-I BD” authors found a significant association between hyperthymic temperament and manic disease onset [16]. Nevertheless, until now, no investigation has investigated the influence of premorbid affective temperaments on the polarity of disease onset (and vice versa) in a sample encompassing both “type-I BD” and “type-II BD” patients. Affective temperaments may be variably associated with different polarities of disease onset in both subtypes of BD, subsequently influencing the clinical course of the illness.

The purpose of the present study is to bridge this knowledge gap by examining the relationship between the polarity at disease onset, temperament, and the clinical characteristics of the illness in a large cohort of individuals diagnosed with both “type-I BD” and “type-II BD.” We hypothesize that the polarity observed at disease onset will serve as an indicator of the subsequent course of BD. Additionally, we expect that the mood polarities observed at the onset of the illness will align with the corresponding premorbid affective temperament in individuals with BD.

## 2. Materials and Methods

### 2.1. Participants

Outpatients with DSM-5 diagnoses of “type-I BD” (BD-I) and “type-II BD” (BD-II) were recruited at the Psychiatry Department of the Fondazione Policlinico Universitario Agostino Gemelli IRCCS in Rome, Italy. Diagnosis was confirmed using the Structured Clinical Interview for DSM-5 [17]. Diagnostic interviews were conducted by trained assessors with demonstrated high interrater reliability (k = 0.87). By implementing stringent inclusion and exclusion criteria, the study aimed to ensure a representative and reliable sample for the investigation. In addition to a DSM-5 diagnosis of BD, inclusion criteria were as follows: (a) age 18–65 years; (b) at least 5 years of education; (c) fluency in Italian; (d) at least 6 months of stable pharmacotherapy for BD. Exclusion criteria were: (a) a history of psychosis unrelated to the primary mood disorder; (b) traumatic brain injury with loss of consciousness; (c) major medical or neurological conditions; (d) a Mini-Mental State Examination (MMSE) score [18] of less than 24 (since scores below this level indicate cognitive deterioration based on normative data from the Italian population); (e) current substance use disorder. Based on the above inclusion/exclusion criteria, we enrolled 191 patients in this study. The sample size was considered appropriate based on previous studies in the field [19,20].

The study conformed to the Principles of Human Rights, as adopted by the World Medical Association at the 18th WMA General Assembly in Helsinki, Finland, in June 1964 and subsequently amended at the 64th WMA General Assembly in Fortaleza, Brazil, in October 2013. All participants gave written informed consent to participate in the study after receiving a full explanation of the study procedures and objectives. Patients received no financial compensation for this study. The commitment to ethical standards, as reflected in adherence to the World Medical Association’s principles, underscored the importance of safeguarding participant rights and well-being throughout the research process. The thoroughness of diagnostic procedures and ethical safeguards contributes to the robustness and reliability of the study’s findings, enhancing the scientific and ethical integrity of the research endeavor. The study was approved by local ethics committees.

### 2.2. Assessment

A semi-structured interview, employed in prior studies [21], was utilized for comprehensive data collection on anamnestic characteristics and clinical information, with a specific focus on the polarity at disease onset (PO). Administered by an experienced psychiatrist, this interview adhered to DSM criteria and clinical assessments, steering clear of simplistic yes/no responses to ensure nuanced insights. Question wording was adaptable for clarity, and the final evaluation incorporated inputs not only from the patients but also from family members/close friends (who were consistently present for at least one visit) and relevant medical records.

All gathered data, spanning family history, psychiatric background, and current psychiatric status, were meticulously entered into preprinted medical records. Following established conventions [8], patients were classified as having a “depressive PO” (D-PO) if the initial episode was depressive, and as “manic/hypomanic PO” (M-PO) if the first-occurring episode was manic or hypomanic. The utilization of a hetero-administered interview approach, coupled with the inclusion of collateral information from various sources, contributes to the robustness and depth of the collected data, enhancing the validity of the study’s findings.

Affective temperaments (cyclothymic, depressive, irritable, hyperthymic, and anxious) were assessed using the short, 39-item version of the validated Italian Temperament Evaluation of Memphis, Pisa, Paris and San Diego—Auto-questionnaire (TEMPS-A-SV) [22]. This instrument is widely used in research and has shown good psychometric properties and optimal factor structure [23]. The original TEMPS-A scale comprises a total of 110 items, with each of the five temperament dimensions represented by about 20 items each. For our study we used the shorter version, validated in Italian, known as the TEMPS-A Short Version (TEMPS-A-SV), which includes a subset of items from the full scale [22]. The TEMPS-A-SV is designed for a more time-efficient assessment while still capturing essential information about an individual’s temperament. This shortened version consists of 39 items, providing a streamlined yet effective tool for evaluating the five temperament dimensions. The distribution of items across the dimensions in the TEMPS-A-SV is generally proportional to the full version, allowing for a quick but reliable analysis of cyclothymic, hyperthymic, depressive, irritable, and anxious traits. This abbreviated version is often employed in settings where a more concise evaluation is needed, making it a versatile option for both research and clinical purposes [22].

The test–retest reliability of the TEMPS-A ranged from 0.58 for the irritable, to 0.68 for the cyclothymic, to 0.69 for the dysthymic, and 0.70 for the hyperthymic temperament in the valedictory study [24]. The instrument showed an excellent internal consistency, with Cronbach’s α ranging from 0.76 for the dysthymic to 0.88 for the cyclothymic temperament in the same study. In another study with the short version, Cronbach’s α was 0.72 for the cyclothymic, 0.71 for the depressive, 0.69 for the irritable, 0.54 for the hyperthymic, and 0.62 for the anxious temperament, while for the entire construct it was 0.80 [25]. For the TEMPS-A-SV, Cronbach’s α was 0.79 (95% Confidence Intervals (C.I.s) from 0.76 to 0.82) for the cyclothymic, 0.72 (95% C.I.s from 0.68 to 0.76) for the depressive, 0.72 (95% C.I.s from 0.68 to 0.76) for the irritable, 0.75 (95% C.I.s from 0.71 to 0.78) for the hyperthymic, and 0.71 (95% C.I.s from 0.66 to 0.75) for the anxious temperament [22]. Reliability estimates were 0.93 for the cyclothymic, 0.92 for the irritable and the depressive, 0.91 for the hyperthymic, and 0.87 for the anxious temperament [22].

### 2.3. Statistical Analyses

To meet our objectives, we divided our sample into two groups: patients with D-PO and patients who reported M-PO. Variables were reported as percentages or means ± SD as appropriate. We initially tested differences in sociodemographic and clinical characteristics across the two groups using standard univariate/bivariate comparisons of continuous measures (ANOVA) and categorical measures (contingency table/χ^2^). The same type of analysis has been used to assess differences in TEMPS-A-SV mean values between the two groups. Mean differences were reported using Cohen’s d as effect size measure.

In addition, we regressed all the factors that were significantly associated with D-PO and M-PO in bivariate analyses on PO, in a multivariate logistic regression with PO as the dependent outcome measure, together with age and sex, in order to consider demographic differences [26].

Where appropriate, we investigated possible multicollinearity between the variables of interest using the variance inflation factor (VIF) indicator obtained from linear regression analysis. Analyses were performed using the statistical routines of SPSS Statistics 24.0 for Windows (IBM Co., Armonk, NY, USA). All statistical tests used a significance level of *p* < 0.05.

## 3. Results

In the total group of patients, 101 (52.9%) had depressive polarity (D-PO) at onset and 90 (47.1%) had manic/hypomanic polarity (M-PO) at onset. The sociodemographic and clinical characteristics of the sample are shown in Table 1.

### 3.1. Univariate and Bivariate Analyses

Univariate/bivariate analyses revealed that patients with D-PO and M-PO differed significantly in the following clinical characteristics: Bipolar Disorder subtypes, age at first depressive episode, age at first manic/hypomanic episode, illness duration, number of past depressive episodes, seasonality, lifetime suicide risk, lifetime substance use, use of lithium and benzodiazepines. Also, they differed in the presence of dysthymic and hyperthymic temperament (Table 2, Figure 1).

Specifically, patients with D-PO were more frequently diagnosed as “type-II BD” (d = 0.49), reported a younger age at first depressive episode (d = 0.32), longer duration of illness (d = 0.34), more past depressive episodes (d = 0.54), higher lifetime suicide risk (d = 0.30), more benzodiazepine use (d = 0.49), and higher scores for dysthymic temperament (d = 0.40).

Conversely, patients with M-PO were frequently diagnosed as “type-I BD” (d = 0.49), reported a younger age at first manic/hypomanic episode (d = 0.41), reported greater seasonality (d = 0.34), lifetime substance use (d = 0.31), higher lithium use (d = 0.31), and higher scores for hyperthymic temperament (d = 0.53).

### 3.2. Multivariate Logistic Regression

Multivariate logistic regression revealed that “type-II BD” (Wald = 5.60; *p* = 0.01 OR: 3.39; 95% Confidence Interval [95%CI]: 1.23–9.35) and age at first depressive episode (Wald = 22.28; *p* < 0.001; OR: 0.41; 95%CI:0.29–0.60) were associated with D-PO, whereas “type-I BD” age at first manic/hypomanic episode (Wald = 21.72; *p* < 0.001, OR: 0.46; 95%CI: 0.33–0.64), and hyperthymic temperament (Wald = 8.12; *p* = 0.004; OR: 1.41; 95%CI: 1.11–1.79) were associated with M-PO (Table 3). The model explained 69% (Nagelkerke R^2^) of the variance in PO. There was no significant multicollinearity, as indicated by the fact that the VIF of the variables of interest was <2.

## 4. Discussion

This is the first study to examine the relationship between temperament and polarity at disease onset (PO) as a clinical course specifier of BD. The results confirmed our original hypothesis; we showed that PO can impact the clinical course of BD and is influenced by premorbid affective temperament. Specifically, temperament seems to align with the type of episode that initially occurs in bipolar disorder. Consistently, we observed a higher frequency of depressive onset (D-PO) in patients with a dysthymic temperament and a more frequent manic/hypomanic onset (M-PO) in patients with a hyperthymic temperament. Notably, multivariate analyses underscore hyperthymic temperament as a significant risk factor for M-PO.

These findings align with initial observations indicating a positive correlation between the number of manic episodes in BD and hyperthymic temperament, while depressive episodes were linked to dysthymic temperament [25]. Further studies have substantiated this relationship, particularly within the framework of the “predominant polarity” construct developed by Eduard Vieta’s group [27]. This construct delineates the prevailing direction of mood episodes throughout an individual’s illness history, indicating a predominance of either manic or hypomanic episodes, or depressive episodes. Different predominant polarities correlate with distinct clinical characteristics. With respect to temperamental traits, Azorin and colleagues noted that patients with predominantly manic recurrences exhibit a stronger hyperthymic temperament compared to those with depressive predominant polarity [28]. In parallel, earlier research has established a specific connection between the construct of predominant polarity and PO, suggesting that the direction of the first episode in the illness often aligns with the more frequently occurring episode direction [29,30]. In our study, we build upon these observations, indicating a potential association between temperament, PO, and the predominant polarity of the impending illness. This highlights a risk continuum from temperamental traits to the clinical course of BD.

Our data may also align with recent biological findings in BD. A recent review highlighted that polarity at onset is among the familial traits observed across generations of BD patients [31]. The study specified that individuals with BD were more likely to exhibit the same polarity of mood episodes (i.e., depressive, manic, or mixed) at the onset of illness as their affected relatives. In parallel, previous evidence has underscored that all temperament theories presume a biological basis for individual differences, with moderate genetic influences demonstrated in twin and adoption studies [32]. Given the enduring nature of temperament as a “trait” and its stability across the lifespan, the association we demonstrated with polarity at onset in patients with BD could serve as a link between phenotypic presentation and genetic vulnerability. However, further longitudinal studies are essential to elucidate and confirm this initial observation.

Our data confirmed that PO influences the clinical course of BD and suggest the existence of two distinct “phenotypes” of the disease: one with D-PO, dysthymic (depressive) temperament, “type-II BD” diagnosis, younger age at first depressive episode, longer duration of illness and higher suicide risk; the other with M-PO, hyperthymic temperament, “type-I BD” diagnosis, younger age at first manic/hypomanic episode, higher seasonality, more frequently treated with lithium. These results are in line with what emerged from a recent large study on the characterization of patients with BD, that confirmed the clinical relevance of disease onset phenotypes [33].

In our sample, depressive onset emerged as the most prevalent phenotype, constituting at least 50% of the initial presentations, a pattern consistent with prior research [5,6,7,8]. Clinical features associated with depressive onset (D-PO) align with the existing literature, encompassing a higher frequency of depressive episodes and an extended duration of illness [ibidem]. These findings echo earlier studies emphasizing that the clinical course of bipolar disorder (BD) tends to be less severe and recurrent in patients primarily experiencing mania/hypomania compared to those primarily presenting with depression [21,31]. They underscore the importance of early prevention strategies for patients presenting with depressive symptoms [32,34].

In line with previous reports [29,33,35], our study also identified an elevated risk of suicide in patients with D-PO. This observation may indirectly correlate with other studies where a dysthymic (depressive) temperament was identified as a risk factor for suicide, while a hyperthymic temperament was deemed protective [36]. The higher prevalence of benzodiazepine use in the D-PO group might be explained in relation to temperament, suggesting that the use of anxiolytics could be an attempt by patients to manage depressive or anxious temperamental traits, as previously proposed [37].

The second clinical phenotype identified in this study involved patients who initially presented with mania/hypomania. Results revealed that individuals with manic/hypomanic onset (M-PO) reported higher lifetime substance use and greater use of lithium. The former is unsurprising, given the common occurrence of substance use during manic/hypomanic episodes [30], which also aligns with the prescription of lithium. Additionally, long-term responsiveness to lithium has been positively associated with the hyperthymic temperament and negatively associated with other temperament types [34]. Furthermore, our findings indicated that patients with M-PO reported higher seasonality than those with depressive onset (D-PO). This contrasts with previous research suggesting that patients with depressive onset are more susceptible to seasonal changes [38]. Nevertheless, our results are in line with recent data indicating that patients with BD-II exhibit less seasonality than those with BD-I [1]. In our sample, patients with BD-I predominantly had M-PO, while those with BD-II had D-PO [39], supporting the consistency of our results. Additional studies are required to further elucidate this point.

Our study showed a strong relationship between PO and BD subtypes, with D-PO associated with “type-II BD” and M-PO associated with “type-I BD”. This is in line with Daban and colleagues, who specifically aimed to define the clinical profile of both “type-I BD” and “type-II BD” using PO [8], and with very recent data from Brancati and colleagues [36]. Our findings are consistent with those of authors showing that BP-I tends to have more manic episodes, whereas BP-II tends to be more chronic and depressive episodes predominate [29]. Multivariate analyses also showed that patients with D-PO are younger at their first depressive episode, while patients with M-PO are younger at their first manic/hypomanic episode. This is further strengthened by the finding that first depressive recurrences occur earlier in BD-II, whereas first (hypo)manic recurrences occur earlier in BD-I [40]; so it appears that the course of BD can be predicted by the direction of its polarity at the outset, with age being a moderator of the course. Furthermore, episodes at a lower age at the beginning of BD bear the same sign that could be expected from belonging to the BD-I or to the BDI-II diagnostic group. Interestingly, in our sample, the D-PO group reported their first manic episode on average 7.67 ± 1.7 years after their first depressive episode. These results confirm that the BD diagnosis must be considered even when patients present with a depressive episode that is not immediately followed by mania/hypomania. Previous studies have shown that the interval between the first depressive episode and mania/hypomania can be very long and varies greatly from patient to patient [41,42]. This variability could be explained by the delay in BD diagnosis, which is mainly due to the difficulty in recognizing mania/hypomania compared to depression, and its consequent treatment with antidepressants. The difficulty, rather than the ability of a given physician to diagnose each condition, is likely due to the tendency of patients to refer to the healthcare system. In fact, while patients feeling depressed are likely to consult a physician, those feeling elated are unlikely to do so. Accordingly, an average of 8 years elapses between a patient’s first episode and a correct BD diagnosis [8]. On the other hand, in the M-PO group, the first depressive episode occurs on average 1.03 ± 0.37 years after the first manic episode. This seems to confirm the hypothesis of the primacy of mania proposed by Athanasios Koukopoulos, according to which depression is a consequence of mania [43,44,45]. “Mania is the fire and depression is its ashes”, he used to state frequently. Koukopoulos’ primacy of mania hypothesis received biological support [45] and has considerable implications for what concerns the use of antidepressant medications in the depressive phase of BD [46]. In fact, the current trend is to provide a basis of mood stabilization before adding an antidepressant for bipolar depression [47,48].

When it comes to treatment implications, we have to bear in mind that a depressive polarity onset coupled to depressive, irritable, cyclothymic, and anxious temperaments may expose patients to a switch to the opposite polarity when exposed to antidepressants, and that the latter should always be administered with mood stabilizers [48]. Conversely, in patients experiencing manic polarity onset, a combination of antipsychotic medications and mood stabilizers is frequently required to effectively address the symptoms present during the initial stages of the illness [49].

Before presenting our conclusions, we must point out some issues that might limit the generalizability of our results. The cross-sectional nature of our study limits our ability to confirm our predictions. Further longitudinal studies are needed to extend our original speculations. In addition, the retrospective nature of the study may have led to uncontrolled recall bias. To minimize the risk of recall bias, we adopted a detailed semi-structured interview based not only on patient reports but also on information from family members and close friends (who were present for at least one visit) and on all available medical records. Finally, one of the defects of our approach was that we classified initial episodes as either depressive or manic/hypomanic and failed to categorize episodes with mixed features as mixed, but we rather pooled them along the manic ones. This represents an inherent limitation of polarity at onset (PO) as a course specifier, as outlined in the original construct [8].

## 5. Conclusions

Summarizing our evidence, we have shown that differences in temperament predisposition, especially in the hyperthymic affective temperament trait, influence the polarity of the first-occurring episode of BD. Using a large sample of patients with either “type-I BD” or “type-II BD”, we confirmed our initial hypothesis that PO is congruent with the respective premorbid affective temperament. Accordingly, the systematic assessment of premorbid affective temperaments may play a crucial role in confirming whether the initial episodes of the disorder were characterized by mania, hypomania (elevated mood), or depression. These findings hold the potential to validate the subtyping of BD based on first-episode polarity and can contribute significantly to tailoring specific prevention and treatment strategies. Future studies could employ both polarity at onset and affective temperaments and be longitudinal in design.

In addition, our findings on the effects of PO on the clinical course of BD point to the need for improved prevention and treatment systems. In patients with depressive onset, who often present with a higher suicidal risk, different pharmacological treatments may be considered compared to those with manic/hypomanic onset, who tend to exhibit more pronounced seasonality and substance use. Tailoring treatments based on these distinct clinical characteristics can enhance the effectiveness and safety of pharmacological interventions, thereby addressing the specific needs of each patient subgroup and avoiding possible side effects. Furthermore, our study underscores the significance of simultaneously considering both “type-I BD” and “type-II BD”. We highlight that these subtypes exhibit distinct clustering patterns based on polarity at disease onset, and accordingly they manifest varying clinical characteristics.

Neurobiological studies could further help delineate subtypes of patients according to PO, with the goal of refining early intervention strategies and better characterizing the heterogeneity of BD.

## Figures and Tables

**Figure 1 brainsci-14-00017-f001:**
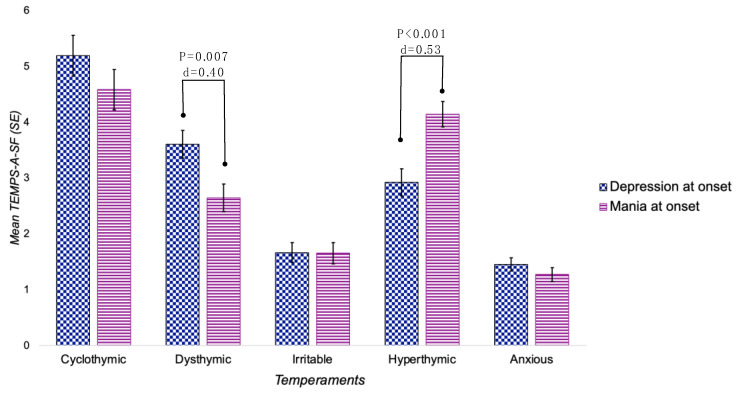
Histograms showing differences in TEMPS-A-SV scores (mean ± SE) according to polarity at onset. *p*, significance level; d, Cohen’s d (effect size).

**Table 1 brainsci-14-00017-t001:** Sociodemographic and clinical characteristics of the sample according to polarity onset (*n* = 191).

	Depression at Onset (*n* = 101)	Mania at Onset (*n* = 90)	*F* or *χ*^2^	df	*p*
*Socio-demographic and lifestyle characteristics*					
Age, y—mean ± SD	45.54 ± 13.21	42.42 ± 11.52	2.99	1	0.085
Gender, males—*n* (%)	41 (40.6)	49 (54.4)	3.66	1	0.056
Married partner—*n* (%)	50 (49.5)	39 (43.3)	0.73	1	0.393
Children—*n* (%)	56 (55.4)	40 (44.4)	2.30	1	0.129
Smoking, yes—*n* (%)	46 (45.5)	46 (51.1)	0.59	1	0.442
Substance use, lifetime—*n* (%)	23 (22.8)	33 (36.7)	4.43	1	**0.035** *
*Clinical variables*					
Diagnostic status—*n* (%)			10.92	1	**0.001 ****
BD I	48 (47.5)	64 (71.1)			
BD II	53 (52.5)	26 (28.9)			
Age first depressive episode, y—mean ± SD	27.20 ± 11.86	30.77 ± 10.60	4.37	1	**0.038 ***
Age first manic episode, y—mean ± SD	34.87 ± 13.56	29.74 ± 10.97	7.81	1	**0.006 ****
Duration of illness, y—mean ± SD	17.30 ± 11.97	13.58 ± 9.98	5.36	1	**0.022 ***
Past depressive episodes—mean ± SD	6.36 ± 6.21	3.63 ± 3.37	13.71	1	**<0.001 *****
Past manic/hypomanic episodes—mean ± SD	5.53 ± 6.70	4.74 ± 4.41	0.90	1	0.343
Family history of psychiatric disorders—*n* (%)	72 (71.3)	62 (68.9)	0.13	1	0.718
Seasonality—*n* (%)	33 (32.7)	44 (48.9)	5.20	1	**0.023 ***
Switch—*n* (%)	38 (37.6)	29 (32.2)	0.61	1	0.435
Hospitalizations—*n* (%)	53 (52.5)	59 (65.6)	3.36	1	0.067
Suicidality—*n* (%)	68 (67.3)	48 (53.3)	3.91	1	**0.048 ***
*Medications use*					
Antipsychotics—*n* (%)	64 (63.4)	56 (62.2)	0.27	1	0.870
Antiepileptics—*n* (%)	56 (55.4)	48 (53.3)	0.086	1	0.770
Lithium—*n* (%)	51 (50.5)	59 (65.6)	4.42	1	**0.036 ***
Benzodiazepines—*n* (%)	53 (52.5)	26 (28.9)	10.92	1	**0.001 ****

* *p* < 0.05; ** *p* < 0.01; *** *p* < 0.001; df, degrees of freedom; *p*, significance level; SD, standard deviation; *n*, number of observations; y, years.

**Table 2 brainsci-14-00017-t002:** Differences in TEMPS-A-SV scores (mean ± SD) according to polarity at onset.

	Depression at Onset (*n* = 101)	Mania at Onset (*n* = 90)	*F* or *χ*^2^	df	*p*
Cyclothymic	5.19 ± 3.67	4.58 ± 3.41	1.41	1	0.237
Dysthymic	3.60 ± 2.48	2.64 ± 2.37	7.43	1	**0.007 ****
Irritable	1.66 ± 1.78	1.65 ± 1.83	0.00	1	0.996
Hyperthymic	2.92 ± 2.41	4.14 ± 2.14	13.65	1	**<0.001 *****
Anxious	1.45 ± 1.21	1.27 ± 1.17	0.96	1	0.329

** *p* < 0.01; *** *p* < 0.001; df, degrees of freedom; *p*, significance level; SD, standard deviation; TEMPS-A-SV, Temperament Evaluation of Memphis, Pisa, Paris and San Diego—Auto-questionnaire, 39 item.

**Table 3 brainsci-14-00017-t003:** Multivariate logistic regression analyses.

	OR	95%CI	Wald	*p*
D-PO				
Type-II BD	3.39	1.23–9.35	5.60	**0.01 ***
Age at first depressive episode	0.41	0.29–0.60	22.28	**<0.001 *****
M-PO				
Type-I DB	3.39	1.23–9.35	5.60	**0.01 ***
Age at first manic/hypomanic episode	0.46	0.33–0.64	21.72	**<0.001 *****
Hyperthymic temperament	1.41	1.11–1.79	8.12	**0.004 ****

* *p* < 0.05; ** *p* < 0.01; *** *p* < 0.001; df, degrees of freedom; *p*, significance level; SD, standard deviation; TEMPS-A-SV, Temperament Evaluation of Memphis, Pisa, Paris and San Diego—Auto-questionnaire, 39 item.

## Data Availability

The data presented in this study are available on request from the corresponding author. The data are not publicly available due to data availability, all used data are available after request to the authors, because they are stored in protected datasets in our institution.

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
