# Peer review of "What Came First, Mania or Depression? Polarity at Onset in Bipolar I and II: Temperament and Clinical Course"

_brainsci, 2023, doi:10.3390/brainsci14010017_

Round 1

Reviewer 1 Report

Comments and Suggestions for Authors

Upon reviewing the paper titled "Who came first, mania or depression? Polarity at onset in Bipolar I and II, temperament and clinical course", which explores the influence of temperament on the polarity at onset (PO) in bipolar disorder (BD), it is evident that while the study addresses an interesting aspect of BD, there are significant shortcomings that undermine its contribution to the field. The paper lacks innovative findings or significant new insights into the relationship between affective temperament and BD PO. Although the study attempts to correlate temperament with the type of first episode in BD patients, the results do not offer substantial advancements beyond what is already known in the field.

Furthermore, the methodology, particularly the statistical analysis, seems to follow standard univariate/bivariate methods without introducing novel or more sophisticated approaches that could provide deeper insights. The paper also suffers from a lack of visualization of the data, making it difficult to grasp the nuances and significance of the findings effectively.

The conclusions drawn appear to be somewhat generic, emphasizing the importance of assessing temperament in BD patients without providing concrete, actionable insights or novel therapeutic directions. Due to these limitations and the lack of significant new contributions to the domain of bipolar disorder research, the paper may not warrant publication in its current form. More in-depth research, possibly incorporating innovative methodologies or exploring uncharted aspects of the relationship between temperament and BD, would be necessary to make a meaningful contribution to the field.

Comments on the Quality of English Language

Changes required. 

Author Response

Upon reviewing the paper titled "Who came first, mania or depression? Polarity at onset in Bipolar I and II, temperament and clinical course", which explores the influence of temperament on the polarity at onset (PO) in bipolar disorder (BD), it is evident that while the study addresses an interesting aspect of BD, there are significant shortcomings that undermine its contribution to the field. The paper lacks innovative findings or significant new insights into the relationship between affective temperament and BD PO. Although the study attempts to correlate temperament with the type of first episode in BD patients, the results do not offer substantial advancements beyond what is already known in the field.

We sincerely appreciate the reviewer’s feedback on our study. While we regret that he/she may not have found it particularly innovative, we would like to highlight that, to the best of our knowledge, there are no studies surpassing ours in examining the correlation between onset polarity and affective temperaments, considering both bipolar disorder type I and II. It's important to note that only one prior study in the field explored the relationship between temperament and polarity at disease onset. Nevertheless, it specifically concentrated on bipolar disorder type I. Our study represents a pioneering effort, as it is the first to comprehensively examine the interplay between polarity at disease onset, temperament, and the clinical course of illness in a substantial sample of patients with both "type-I BD" and "type-II BD." This is clearly stated in the introduction section.

Furthermore, the methodology, particularly the statistical analysis, seems to follow standard univariate/bivariate methods without introducing novel or more sophisticated approaches that could provide deeper insights. The paper also suffers from a lack of visualization of the data, making it difficult to grasp the nuances and significance of the findings effectively.

We express our gratitude to the reviewer for drawing attention to the need for data visualization. In response to this valuable suggestion, we have incorporated Figure 1 into the manuscript.

Regarding the statistical methods, we want to clarify that we employed more than simple univariate/bivariate analyses. Our approach involved conducting univariate/bivariate analyses followed by the application of multivariate logistic regression models. It is crucial to highlight that we diligently assessed the potential for multicollinearity in the multivariate model. Notably, the regression model demonstrated both statistical significance and remarkable robustness, explaining 69% of the variance according to Nagelkerke R2.

We would like to underscore that this type of modeling aligns with current practices in numerous studies featured in high-impact factor journals, exemplified by works such as Tundo et al., Eur Psychiatry 2023 (PMID: 37697671), and Moccia et al., Brain Behav Immun. 2020 (PMID: 32325098).

The conclusions drawn appear to be somewhat generic, emphasizing the importance of assessing temperament in BD patients without providing concrete, actionable insights or novel therapeutic directions. Due to these limitations and the lack of significant new contributions to the domain of bipolar disorder research, the paper may not warrant publication in its current form. More in-depth research, possibly incorporating innovative methodologies or exploring uncharted aspects of the relationship between temperament and BD, would be necessary to make a meaningful contribution to the field.

According to the reviewer’ s observation we expanded the conclusions section, focusing on actionable insights. We thank the reviewer for his/her thoughtful review of our paper. While we respect his/her perspective, we believe that our study does provide valuable insights into the relationship between polarity at onset, temperament, and the clinical course of bipolar disorder. We also sincerely think that the reviewer’s observations significantly contribute to improve the quality of the manuscript.

Reviewer 2 Report

Comments and Suggestions for Authors

This study explored the influence of the first mood episode in bipolar disorder (BD) on the clinical evolution in these patients, and the relationship between affective temperaments and the nature of polarity onset. The topic is interesting, and the results may be relevant for clinicians working with patients diagnosed with BD. Please refer to the following observations:

Lines 39-40 – Which previous studies? Please insert references for this sentence;

Throughout the manuscript, „BD type I (or II)” would be better replaced by „type I (or II) BD”;

Lines 135-140- Were substance use disorders used as exclusion criterion? The evolution of BD may be significantly influenced by addictive comorbidities.

Line 195- a comma is needed instead of a semicolon;

Maybe considering including a distinct chapter for „Conclusions” (lines 355-362).

Any suggestions for future research based on the limitations of the current study or aspects identified during this search?

Comments on the Quality of English Language

Minor editing of English language recommended.

Author Response

This study explored the influence of the first mood episode in bipolar disorder (BD) on the clinical evolution in these patients, and the relationship between affective temperaments and the nature of polarity onset. The topic is interesting, and the results may be relevant for clinicians working with patients diagnosed with BD.

We thank you for finding our paper interesting for clinicians working with BD patients.

Please refer to the following observations:

Lines 39-40 – Which previous studies? Please insert references for this sentence; 

We thank you for this observation; we added references. They were the same as the ones we cited a little further on.

Throughout the manuscript, „BD type I (or II)” would be better replaced by „type I (or II) BD”;

We replaced. Thank you for your observation.

Lines 135-140- Were substance use disorders used as exclusion criterion? The evolution of BD may be significantly influenced by addictive comorbidities.

We thank the reviewer for his thoughtful observation. Yes, current substance use disorder was an exclusion criterion. We added this information.

Line 195- a comma is needed instead of a semicolon;

We thank you; it was a misprint. We corrected.

Maybe considering including a distinct chapter for „Conclusions” (lines 355-362).

We thank you for this suggestion. We put a fifth subheading to summarize our last paragraph.

Any suggestions for future research based on the limitations of the current study or aspects identified during this search?

We added suggestions in the Conclusions paragraph. We thank you for thoughtful suggestions that helped our paper to improve considerably.

Reviewer 3 Report

Comments and Suggestions for Authors

The manuscript addresses an interesting topic about mania and depression

In general, the manuscript is organized and well-written, however I have the following concerns

·        How did the authors came up with the sample size? Was there any predetermined calculations, etc…

·        Were the scales used all in Italian language? Were they self-administered or administered by the doctors

·        The study is basically an association study, although the regression analysis were carried out, however why there were no tables for them, why the reporting was not based on Odds ratios and the 95 %CI?

Author Response

The manuscript addresses an interesting topic about mania and depression

    We thank you for finding our paper interesting.

In general, the manuscript is organized and well-written, however I have the following concerns

  • How did the authors came up with the sample size? Was there any predetermined calculations, etc…

      We thank the reviewer for his observation. We did not apply a sample power calculation to determine the sample size. Patients were consecutive enrolled for the study. Sample size was considered appropriate based on previous studies in the field (Please see PMID: 34990627; PMID: 36566232). This is now specified in the manuscript.

  • Were the scales used all in Italian language? Were they self-administered or administered by the doctors

Affective temperaments (cyclothymic, depressive, irritable, hyperthymic, and anx-ious) were assessed using the short, 39-item version of the validated Italian Temperament Evaluation of Memphis, Pisa, Paris and San Diego-Autoquestionnaire (TEMPS -A-39). This is clearly stated in the method section.

          The study is basically an association study, although the regression analysis were carried out, however why there were no tables for them, why the reporting was not based on Odds ratios and the 95 %CI?

As specified in the methods section, univariate/bivariate analyses were followed by multivariate regression analyses. For multivariate logistic regression, in addition to p value, we reported Wald, demonstrating the significance of individual coefficients in the model. Please note that the regression model resulted significant and particularly robust (The model explained 69%, of the variance according to Nagelkerke R2). Following reviewer’s suggestion, we also added Odds ratio and the 95%CI (please see Results section).

We thank you for useful suggestions that rendered our paper a better one.

Round 2

Reviewer 1 Report

Comments and Suggestions for Authors

While the paper provides some insights into the relationship between affective temperaments, the polarity at onset (PO), and the clinical course of bipolar disorder (BD), many technical issues merit a drastic change or major revisions for clarity and rigor:

1. The introduction lacks a clear statement of the research gap or question that the study aims to address. Provide a more explicit background on why understanding the relationship between affective temperaments and PO is crucial for advancing knowledge in the field.

2. Methods: Specify the inclusion and exclusion criteria for selecting the 191 patients with BD. This is crucial for readers to assess the generalizability of the findings.

Provide more details on the short version of the Temperament Evaluation of Memphis, Pisa, Paris, and San Diego-Autoquestionnaire (TEMPS -A-39-SV) to ensure transparency and reproducibility of the assessment.

3.Elaborate on the statistical methods used for univariate/bivariate analyses and multivariate logistic regression models. Readers need a more detailed understanding of the analytical approach to assess the robustness of the findings.

4. The results section lacks a clear structure. Consider organizing the findings in a more systematic manner, possibly using tables or figures for better visual representation of the data.

5.Provide effect sizes or confidence intervals for significant associations to help readers gauge the clinical significance of the observed relationships.

6. While the conclusion emphasizes the importance of assessing temperament and PO, the discussion should delve deeper into the clinical implications of the findings. How can this knowledge impact treatment decisions, and what are the potential practical applications?

7. Include a comprehensive discussion of the limitations of the study, such as potential biases or confounding variables. Additionally, it suggest avenues for future research to address these limitations and further advance the understanding of the relationship between affective temperaments and BD.

8.Ensure consistency in terminology throughout the paper. For instance, use a consistent format for referring to the TEMPS questionnaire (e.g., TEMPS-A-39-SV) to avoid confusion.

9. Enhance the abstract by providing a succinct summary of the study's objectives, methods, key findings, and implications. This will help readers quickly grasp the significance of the research.

The paper can achieve clarity, transparency, and impact by addressing these technical issues. 

Comments on the Quality of English Language

Needs to be edited

Author Response

We respond point-to-point here underneath each point raised; the revisor can find changes with respect to the initial version in red text.

While the paper provides some insights into the relationship between affective temperaments, the polarity at onset (PO), and the clinical course of bipolar disorder (BD), many technical issues merit a drastic change or major revisions for clarity and rigor:

  1. The introduction lacks a clear statement of the research gap or question that the study aims to address. Provide a more explicit background on why understanding the relationship between affective temperaments and PO is crucial for advancing knowledge in the field.

We thank reviewer for this observation. We clarified our statements in introduction about the clinical importance of affective temperaments, as well as why we chose PO as an outcome measure. We further clarified the specific aim of the study and our hypotheses.

  1. Methods: Specify the inclusion and exclusion criteria for selecting the 191 patients with BD. This is crucial for readers to assess the generalizability of the findings.

In Methods section, Participants subsection, we described (line 139-147) as follow: “In addition to a DSM-5 diagnosis of BD, inclusion criteria were as follows: (a) age 1865 years; (b) at least 5 years of education; (c) fluency in Italian; (d) at least 6 months of stable pharmacotherapy for BD. Exclusion criteria were: (a) a history of psychosis unrelated to the primary mood disorder; (b) traumatic brain injury with loss of consciousness; (c) major medical or neurological conditions; (d) a MiniMental State Examination (MMSE) score [18]of less than 24 (since scores below this level indicate cognitive deterioration based on normative data from the Italian population); (e) current substance use disorder. Based on the above inclusion/exclusion criteria, we enrolled 191 patients in this study.” We hope this will address your concerns.

Provide more details on the short version of the Temperament Evaluation of Memphis, Pisa, Paris, and San Diego-Autoquestionnaire (TEMPS -A-39-SV) to ensure transparency and reproducibility of the assessment.

Thank you for the observation; we provided additional details.

3.Elaborate on the statistical methods used for univariate/bivariate analyses and multivariate logistic regression models. Readers need a more detailed understanding of the analytical approach to assess the robustness of the findings.

Thank you for this suggestion; we added details on statistical procedures.

  1. The results section lacks a clear structure. Consider organizing the findings in a more systematic manner, possibly using tables or figures for better visual representation of the data.

We reorganized Results according to your suggestion. We provided a Figure to enhance visual representation of our data. We also added a Table on regression analysis.

5.Provide effect sizes or confidence intervals for significant associations to help readers gauge the clinical significance of the observed relationships.

We provided effect sizes (Cohen’s d) for all significant associations.

  1. While the conclusion emphasizes the importance of assessing temperament and PO, the discussion should delve deeper into the clinical implications of the findings. How can this knowledge impact treatment decisions, and what are the potential practical applications?

We added on the clinical implications of our findings in the Conclusions section.

  1. Include a comprehensive discussion of the limitations of the study, such as potential biases or confounding variables. Additionally, it suggest avenues for future research to address these limitations and further advance the understanding of the relationship between affective temperaments and BD.

Limitations and sources of bias and confounders were addressed in the paragraph prior to Conclusions.

8.Ensure consistency in terminology throughout the paper. For instance, use a consistent format for referring to the TEMPS questionnaire (e.g., TEMPS-A-39-SV) to avoid confusion.

Thank you; we corrected all the non-consistent terminology in the paper.

  1. Enhance the abstract by providing a succinct summary of the study's objectives, methods, key findings, and implications. This will help readers quickly grasp the significance of the research.

We strengthened the Abstract following your suggestion.

The paper can achieve clarity, transparency, and impact by addressing these technical issues. 

We thank you for your suggestions that helped us to improve our paper and we hope that you’ll find it clearer and more precise.